# Immunomodulatory, Antioxidant Activity and Cytotoxic Effect of Sulfated Polysaccharides from *Porphyridium cruentum.* (S.F.Gray) Nägeli

**DOI:** 10.3390/biom11040488

**Published:** 2021-03-24

**Authors:** Virginia Casas-Arrojo, Juan Decara, María de los Ángeles Arrojo-Agudo, Claudia Pérez-Manríquez, Roberto T. Abdala-Díaz

**Affiliations:** 1Instituto de Biotecnología y Desarrollo Azul (IBYDA), Departamento de Ecología y Geología, Facultad de Ciencias, Universidad de Málaga, Facultad de Ciencias, 29071 Málaga, Spain; virginiac@uma.es (V.C.-A.); juandecara@uma.es (J.D.); maarrojo@uma.es (M.d.l.Á.A.-A.); 2Instituto de Investigación Biomédica de Málaga (IBIMA), Hospital Regional Universitario de Málaga, Universidad de Málaga, 29071 Málaga, Spain; 3Departamento de Botánica, Facultad de Ciencias Naturales y Oceanográficas, Universidad de Concepción, Concepción 4030000, Chile; claudiaperez@udec.cl; 4Unidad de Desarrollo Tecnológico, Universidad de Concepción, Concepción 4030000, Chile

**Keywords:** immunomodulation, antioxidant capacity, sulfated polysaccharides, *Porphyridium cruentum*

## Abstract

*Porphyridium cruentum* is a unicellular microalga that can synthesize and secrete to the culture medium-high amounts of polysaccharides. In this study, the immunomodulatory, cytotoxic effect and antioxidant activity of the sulfated polysaccharides (PcSPs) were determinate. The PcSPs were precipitated with 2% Cetylpyridinium bromide hydrate and ethanol and purified by dialysis. The extract was lyophilized for its characterization by Fourier transform-Infrared (FT-IR) spectroscopy and gas chromatography–mass spectrometry (GC-MS). The antioxidant activity of PcSPs were examined with assay 2,2′-azino-bis (3-ethylbenzothiazoline-6-sulfonic acid) and compared with that of the biomass, observing significant differences between the results obtained from the PcSPs and biomass. To determine their ability to induce cytokine production Tumor Necrosis Factor alpha (TNF-α) and interleukina-6 (IL-6), the immunomodulatory activity of the PcSPs has been evaluated. In the mouse macrophage cell line (RAW 264.7), PcSPs are potent inducers of IL-6 cytokines but mainly of TNF-α. The cytotoxic capacity of PcSPs was measured by the MTT colorimetric assay in colorectal carcinoma (HTC-116), human leukemia (U-937 and HL-60), breast cancer (MCF-7), lung cancer (NCI-H460) and human gingival fibroblasts (HGF-1) cell lines. The IC_50_ value of 2311.20 µg mL^−1^, 1676.74 µg mL^−1^, 1089.63 µg mL^−1^, 5498.14 µg mL^−1^ and 2861.49 µg mL^−1^ respectively in the tumor lines and 5022.55 µg mL^−1^ in gingival fibroblasts were obtained. Our study suggested that PcSPs from *P. cruentum* have a moderate immunomodulatory and cytotoxic effect. The results obtained indicate that the polysaccharides from *P. cruentum* are potent inducers of IL-6 cytokines and, most importantly, of TNF-α. PcSPs showed no evidence of antigenic activity or hypersensitivity when administered intraperitoneally in mice. Furthermore, the in vivo study revealed an improvement of local inflammatory response against stress in the peritoneum. These findings suggest that the PcSPs from *P. cruentum* might have potential as a valuable ingredient in nutraceutical products.

## 1. Introduction

Algae are one of the oldest forms of life and constitute a large and versatile group of organisms in terms of size, shape and ecological function [1]. Algae are a source of valuable compounds, with a broad chemical diversity and bioactive properties of interest for pharmaceutical, cosmetic and food industries [2]. For all this, products from marine sources are increasingly valued in the market. Among the metabolites produced by algae, PcSPs offer a broad scope of physiological properties. These metabolites can, for example, be used as an anticoagulant, anti-hyperlipidemic, antiviral or antitumor agents [3,4].

Microalgae have been extensively studied in the last decades because they can be obtained on a large-scale using bioprocesses based on screening, prospecting, cultivation or scaling techniques. These studies made it possible to strengthen the biotechnology industry without affecting the balance of the marine ecosystem, particularly preventing overexploitation of these organisms in their natural habitats [5].

*Porphyridium cruentum* is a unicellular microalga with a cell diameter ranging from 6 to 10 μm [6]. It grows in saline water and can synthesize and secrete high amounts of PcSP to the culture medium [7]. The polysaccharides in its cell walls amounts to more than 50% of its total biomass. These polysaccharides display high antioxidant and anti-inflammatory activity, interfering in the formation and propagation of free radicals and inhibiting several mechanisms that induce or increase the activity of certain enzymes [8,9,10]. The source of these microalgae’s red color is phycoerythrin, which plays a vital role along with other phycobiliproteins present as accessory photosynthetic pigment [11]. Furthermore, *P. cruentum* is also a rich source of lipids and polyunsaturated fats acids [12,13,14], which provide health benefits by reducing the risk of cardiovascular disease, high cholesterol levels [15], inflammation and cancer risk [16].

The polysaccharides in the cell walls of *P. cruentum* are mainly composed of xylose (Xyl) (38%), glucose (Glc) (24%), galactose (Gal) (22%) and glucuronic acid (Gluc-A) (10%), yet arabinose (Arb), rhamnose and mannose (Mann) can also be found in lower concentrations [17,18,19]. Algal cell-wall sugars have been shown to promote biochemical and physiological responses in several vertebrates. Polysaccharides from *P. cruentum*, for example, inhibit viral replication and are a potent hypocholesterolemic agent in rats and chickens [15,20,21,22]. Besides this, these polysaccharides promote anti-oxidant activity suggesting a cell-protecting mechanism against reactive oxygen species [8]. Furthermore, the in vivo administration of the exocellular polysaccharides of *P. cruentum* to mice has resulted in an increase of the macrophage population and an increase of the acid phosphatase enzyme [23,24].

This study assesses the activity of *P. cruentum* as an antioxidant and immunomodulatory agent as well as its cytotoxic effects on different human tumor lines such as human colon cancer (HCT-116), human breast adenocarcinoma (MCF-7), human leukemia (U-937 and HL-60) and human lung cancer (NCI-H460) and in the human gingival fibroblasts cell line (HGF-1). Another imperative objective is to test in vivo, whether the intraperitoneal administration of PcSPs is capable of enhancing the local immune response without triggering any hypersensitivity reactions.

## 2. Materials and Methods

### 2.1. Materials and Chemicals

All samples were stored frozen (−80 °C) and lyophilized prior to use. All chemicals used were of analytical grade and purchased from either Merck (Darmstadt, Germany) or Sigma-Aldrich (St. Louis, MO, USA). All water was obtained from a Milli-Q system (Millipore, Billerica, MA, USA).

### 2.2. Biological Material

A clonal strain of *P. cruentum* (S.F.Gray) Nägeli was provided by the Institute of Biotechnology and Blue Development (IBYDA) of Malaga University (strain code n° UMA-200997). The strain was kept in Vonshak medium [25], and different parameters were measured, keeping the temperature constant at 25 °C, radiation at 100 µmol m^−2^ s^−1^ of photons with a photoperiod of 16 h of light and 8 h of darkness (16/8) and a pH between 6.5 and 7 per periodic aeration with CO_2_. The cultures were kept under agitation using a constant air bubbling system to avoid sedimentation of the algal biomass and to ensure a homogeneous distribution of nutrients and irradiance.

Once the cultures were in the early stationary phase, the biomass was collected by centrifugation at 4500 rpm for 10 min at 4 °C, washed with distilled water and frozen at −80 °C for subsequent lyophilization.

### 2.3. Extraction of Polysaccharides

The extraction of polysaccharides from *P. cruentum* was performed using the method proposed by Abdala-Díaz et al. (2010) [26] with minor modifications. Once the culture of *P. cruentum* reached the early stationary phase, centrifugation was performed at 4500 rpm for 10 min, and the supernatant was recovered for polysaccharides extraction. The polysaccharides were extracted by selective precipitation with 2% Cetylpyridinium bromide hydrate (*w*/*v*) (Cetavlon) (Sigma-Aldrich Ref: 285315). The precipitated sulfated polysaccharides were dissolved with 4 M NaCl (Sigma-Aldrich), centrifuged at 5000 rpm for 10 min and then flocculated with 96% (*v*/*v*) ethanol (Sigma-Aldrich). The final pellet was recovered and introduced into a dialysis membrane (Cellulose dialysis tube, Sigma-Aldrich) for overnight dialysis at low osmotic pressure (1M NaCl) and agitation at 4 °C. After dialysis, the entire membrane content was recovered and washed with 96% EtOH. Final centrifugation (4500 rpm, 2 min, at room temperature) was applied to recover the polysaccharides. The polysaccharides were then frozen at −80 °C in absolute ethanol for 24 h and lyophilized (Cryodos Lyophilizer, Telstar). The sulfated polysaccharide-enriched extract will hereinafter be referred to as “PcSPs”.

### 2.4. Total Carbon (C), Hydrogen (H), Nitrogen (N) and Sulfur (S)

The technique used for the determination of total carbon (C), nitrogen (N) and sulfur (S) in the biomass of dry algae and the polysaccharides extracted from *P. cruentum* was the total combustion used in the LECO TruSppec Micro elemental analyzer CHNSO according to Abdala-Díaz et al. (2019) [27]. The result of each element (C, H, N, S) is expressed in percentages (%) with respect to the sample’s total weight. All determinations were made in triplicate (*n* = 3).

### 2.5. Biochemical Composition of Biomass from P. cruentum

For the determination of the biochemical composition, the assays were carried out with the previously lyophilized biomass. The lipids were determined using the Folch method [28], the total proteins were calculated by multiplying the percentage of total nitrogen by the Lourenço factor N-prot, which, for microalgae in the early stationary phase, is 4.99 [29]. The total carbohydrates were determined according to the method of Dubois [30]. The weight of inorganic compounds was determined using the Ismail method [31]. The humidity level was determined gravimetrically by incineration in a muffle furnace at 550 °C for 3 h. All assays were performed independently and in triplicate (*n* = 3).

### 2.6. Fourier Transform Infrared Spectroscopy (FT-IR)

FT-IR analysis was used to characterize the PcSPs structure and to identify the functional groups in the PcSPs structure [32]. The FT-IR spectra in the 400–4000 cm^−1^ region were obtained using self-supporting pressed disks of 16 mm in diameter of a mixture of polysaccharides and KBr (1% *w*/*w*) with a hydrostatic press at a force of 15.0 tcm^−2^ for 2 min. Thermo Nicolet Avatar 360 IR spectrophotometer (Thermo Electron Inc., Franklin, MA, USA) having a resolution of 4 cm^−1^ with a DTGS detector operated with OMNIC 7.2 software (bandwidth 50 cm^−1^, enhancement factor 2.6). Baseline adjustment was performed using the Thermo Nicolet OMNIC software 9 (Termo Fischer, San Jose, CA, USA). The OMNIC correlation algorithm was used to compare sample spectra with those of the spectral library (Thermo Fischer Scientific).

### 2.7. Gas Chromatography–Mass Spectrometry (GC-MS)

#### 2.7.1. Chemicals and Reagents

The monosaccharides Glc, Gal, Mann, Xyl, apiose, Myo-inositol (Internal Standard, IS), and the pyridine, hexane and methanol/3 M HCl solution were purchased from Sigma Aldrich. Tri-Sil HTP Reagent was purchased from Thermo Scientifics.4.9.2. Derivatization of polysaccharides. TMS (trimethylsilyl) derivatization was performed on previous reports with appropriate modifications [26]. Polysaccharides samples (2 mg) and monosaccharides standards were treated using the same procedure. An aliquot (100 µL) of the standard stock solution of 1 mg mL^−1^ of each monosaccharide was dried under nitrogen gas flow. A mixture of 2 mL MeOH/3M HCl was added to the samples of PcSPs, and a mixture containing the standard monosaccharides. The resultant solution was kept at 80 °C for 24 h [33], subsequently washed with methanol and dried under nitrogen gas flow. The trimethylsilyl reaction was then carried out with 200 µL of Tri-Sil HTP. The sample vials were heated at 80 °C for 1 h. The derivatized samples were cooled to room temperature and dried under a stream of nitrogen. The dry residue was extracted with hexane (2 mL) and centrifuged. Finally, the solution containing silylated monosaccharides was concentrated and reconstituted in hexane (200 µL), filtered and transferred to a GC-MS autosampler vial. Sample preparation and analyses were performed in triplicate.

#### 2.7.2. Gas Chromatography/Mass Spectrometry (GC-MS) Analysis

GC/MS analyses were carried out using a gas chromatography Trace GC (Thermo Scientifics), autosampler Tri Plus and DSQ mass spectrometer quadrupole (Thermo Scientifics). The column was ZB-5 Zebron, Phenomenex (5% Phenyl, 95% Dimethylpolysiloxane) with dimensions of 30 m × 0.25 mm i.d. × 0.25 µm. The column temperature program started at 80 °C (held 2 min) and underwent a gradient of 5 °C/min to reach a final temperature of 230 °C. The carrier gas was helium (flow 1.2 mL/min). The injection volume was 1 µL in splitless mode at 250 °C. The source and MS transfer line temperature were 230 °C. The mass spectrometer was set for a Select Ion Monitoring (SIM) program in electron ionization mode (EI) at 70 eV. The TMS-derivatives were identified by characteristic retention times and mass spectrum compared to those of the standards that were used for the identification of monosaccharides. The compounds were identified by comparing the mass spectra with those in the National Institute of Standards and Technology (NIST 2014) library.

### 2.8. Lipopolysaccharides (LPS) Contamination Assay

The presence of lipopolysaccharides (LPS) in PcSPs fraction isolated from *Porphyridium cruentum* was evaluated using the Limulus amebocyte lysate (LAL) assay kit (Endosafe^®^-PTS, Charles River Laboratories, Charleston, SC, USA) according to the manufacturer guidelines. The amount of endotoxin in the sample was expressed as endotoxin units (EU) mL^−1^.

### 2.9. Antioxidant Capacity as ABTS Assay Scavenging of Free Radical in PcSPs and Biomass

The ability of polysaccharides and algal biomass to remove free radicals was evaluated by an ABTS test according to [34] with few modifications. The ABTS radical (ABTS^+^) was generated by a reaction of 2.2′-Azino-Bis (3-ethylbenzothiazoline-6-sulfonic acid) (ABTS) (7 mM) with of potassium persulfate 2.45 mM. This reaction mixture was stored at room temperature for 16 h. The lyophilized samples (biomass and polysaccharides) were dissolved in phosphate buffer (PB) (4 mg mL^−1^), dilutions were made so that the final concentrations in the cuvettes were 200, 100, 50, 25 and 12.5 µg mL^−1^. Then, 50 µL of each sample was mixed with 940 µL of PB and 10 µL of ABTS solution. The resulting mixture was kept at 20 °C for 8 min to subsequently measure its absorbance at 727 nm against the PB. (±)-6-Hydroxy-2,5,7,8-tetramethylchromana-2-carboxylic acid (Trolox) (Sigma-Aldrich Ref: 238813) was used as a positive control of antioxidant activity.

The radical scavenging capacity of ABTS was calculated according to the Equation (1):AA% = [(A_0_ − A_1_)/A_0_] × 100(1)
where, A_0_ is the absorbance of the ABTS radical in phosphate buffer at time 0; A_1_ is the absorbance of the ABTS radical solution mixed with the sample after 8 min. A calibration curve was plotted using different concentrations of Trolox^®^ (0 to 5 μM), from a stock of Trolox^®^ 2.5 mM. All determinations were performed in triplicate (*n* = 3) [35]. The antioxidant capacity was expressed as % antioxidant activity.

### 2.10. Cell Cultures

The cell lines used in this study were the human colorectal carcinoma (HCT-116, ATCC, Manassas, VA, USA), human breast adenocarcinoma (MCF-7, ATCC, USA), human leukemia (U-937 and HL-60 ATCC, USA) and lung cancer (NCI-H460, ATCC, USA) as tumor lines and human gingival fibroblasts (HGF-1, primary culture, ATCC, USA) as normal cell lines. Cell line murine macrophages (RAW 264.7, ATCC, USA) were used to determine the cytokines. HCT-116, MCF-7, HGF-1 and RAW 264.7 were cultured in Dulbecco’s Modified Eagle’s Medium (DMEM) (Biowest, L0103) supplemented with 10% fetal bovine serum (FBS) (Biowest, South America Origin, S1810) 1% penicillin-streptomycin solution (Biowest, L0022) and 0.5% amphotericin B (Biowest, L0009). The U-937 and NCI-H460 cells were cultured in RPMI-1640 medium (Biowest, L0498) supplemented with 10% FBS, 1% penicillin-streptomycin solution 100× and 0.5% amphotericin B and HL-60 cells were cultured in RPMI-1640 medium supplemented with 20% FBS, 1% penicillin-streptomycin solution and 0.5% amphotericin B. The cells were incubated at 37 °C in humidified air containing 5% CO_2_. When the adherent cells (HTC-116, MCF-7, HGF-1 and NCI-H460) reached a confluence of 75–80%, they were lifted with Trypsin-EDTA 1X in phosphate-buffered saline (PBS) (Biowest, L0940), they were centrifuged at 1500 rpm 5 min at room temperature, and, in the case of the cells in suspension (U-937 and HL-60), were harvested upon reaching 70–75% confluence and centrifuged at 1500 rpm for 5 min at room temperature. Once centrifuged, the corresponding test or the subculture of the cells was carried out.

### 2.11. Cytotoxic Effect Assay

The cytotoxic effect was measured using the MTT assay in six cell lines HCT-116, MCF-7, U-937, HL-60, NCI-H460 and HGF-1. Cells were incubated in 96-well plates independently with different concentrations of PcSPs at 37 °C in a humid atmosphere with 5% CO_2_ for 72 h. The trial was carried out following the method proposed by Abdala-Díaz et al. (2019) [27]. The determinations were carried out in independent experiments in triplicate.

### 2.12. Cell Cycle Analysis by Flow Cytometry

The cell cycle of HL-60 cells was analyzed using the flow cytometry assay by seeding 5 × 10^5^ cells in 6-well plates, the final volume of each well-being 1.5 mL. They were incubated at 37 °C in a humid atmosphere with 5% CO_2_ until sub-confluence. After this, the cells were harvested, centrifuged and cultured again in said plate to be treated with different concentrations of PcSPs, previously dissolved in a fresh and complete culture medium. To carry out the assay, the criterion was to use three concentrations of PcSPs (0.1 mg mL^−1^, 1 mg mL^−1^ and 10 mg mL^−1^), i.e., a concentration 10 times higher than the IC_50_, the concentration IC_50_ and another 10 times below IC_50_. Thus, it was verified how these concentrations of PcSPs affected the HL-60 cell cycle. For the positive control, 2-methoxyestradiol (20 µM) (Sigma-Aldrich, M6383) was used. The plates were incubated under the indicated conditions for 16 h. Cells were harvested and centrifuged. The pellets were washed with PBS and fixed (70% EtOH, 1 h at −20 °C). Finally, the cells were centrifuged and washed twice with PBS, suspended in 40 μg mL^−1^ propidium iodide staining solution (Sigma-Aldrich, P4864) and 0.1 mg mL^−1^ RNase-A (Sigma-Aldrich, R6513) in PBS and incubated for 30 min at 37 °C protected from light. The samples were measured in the FACS VERSETM flow cytometer (BD Biosciences, San Jose, CA, USA), the results were analyzed with the BD FAC-Suite program.

### 2.13. Macrophage Proliferation Assay (RAW 264.7)

For the macrophage proliferation assay (RAW 264.7), 5 × 10^4^ cells in well^−1^ were incubated in the presence of different concentrations of polysaccharides (0–100 μg mL^−1^) in a 96-well microplate for 24 h at 37 °C, 5% CO_2_ and a humid atmosphere. Macrophage proliferation was estimated using the MTT (3-(4,5-dimethylthiazol-2-yl)-2,5-diphenyltetrazolium bromide) method proposed by Abdala-Díaz et al. (2010) [3]. The plate was measured spectrophotometrically at 550 nm (Micro Plate Reader 2001, Whittaker Bioproducts Walkersville, MD, USA). Relative cell viability was expressed as the mean percentage of viable cells compared to untreated cells. Four samples for each concentration tested were included in each experiment. The determinations were carried out in independent experiments in triplicate.

### 2.14. Determination of Cytokines

The RAW 264.7 cells were cultured in 24-well microplates (5 × 10^5^ cells well^−1^) with DMEM at temperature 37 °C, 5% CO_2_, a humid atmosphere and in the presence of different concentrations of polysaccharides (0–75 μg mL^−1^) in a total volume of 1mL. Bacterial LPS from *Escherichia coli* Serotype 055: B5 (Sigma, USA ref: L-4005) (50 ng mL^−1^) was used as a positive control for macrophage activation. The supernatant was collected after 48 h incubation and used to determine cytokine production following the polysaccharides stimulation. The production of TNF-α and IL-6 was measured by sandwich enzyme-linked immunosorbent assays (ELISA) described by Martinez et al. (1998) [36]. Briefly, a purified rat anti-mouse monoclonal TNF-α or IL-6 antibody (0.5 mg, BD Pharmingen) was used for coating at 2 µg mL^–1^ and 4 °C for 16 h. After washing and blocking with PBS containing 3% bovine serum albumin, culture supernatants were added to each well for 12 h at 4 °C. Unbound material was washed off and a biotinylated monoclonal anti-mouse TNF-α or IL-6 antibody (0.5 mg, BD Pharmingen) was added at 2 µg mL^–1^ for 2 h. Bound antibody was detected by the addition of avidin peroxidase (Sigma) for 30 min followed by the addition of the ABTS substrate solution. Absorbances at 405 nm were measured 10 min after the addition of the substrate. A standard curve was constructed using various dilutions of recombinant murine TNF-α or IL-6 in PBS containing 10% fetal calf serum (FCS). The amount of each cytokine in the culture supernatants was determined by the extrapolation of absorbances to the standard curve.

### 2.15. In Vivo Study

#### 2.15.1. Animals and Ethics Statement

The study was carried out in the Animals Experimentation Center of the University of Malaga, performed on 4- to 5-week-old male Balb/C mice weighing 24–26 g (Charles River Laboratories, Barcelona, Spain). Mice were housed in appropriate cages for a maximum of six individuals maintained under a standard 12 h light-dark cycle in a room under temperature/humidity control with ad libitum access to water and food. Experimental procedures with animals were carried out in strict accordance with the recommendations in the European Communities directive 2010/63/EU and Spanish legislation (Real Decreto 53/2013, BOE 34/11370–11421, 2013) regulating the care and use of laboratory animals and making every effort possible to minimize animals suffering and reducing the number of animals used.

#### 2.15.2. Experimental Design and Treatment

PcSPs and LPS were dissolved in a 0.9% saline solution and intraperitoneal (i.p.) administrated at a dose of 10 mg Kg^−1^ of body weight in a volume of 1 mL Kg^−1^ of body weight. The basal group did not receive any administration, while control animals received only 0.9% saline vehicle solution (VEH). The administration of PcSPs, LPS and VEH was carried out in a single dose every three times in the experiment: on days 1, 7 and 14 from the beginning of the study and finally sacrificing the animal’s day 21.

#### 2.15.3. White Blood Cell Differential

A drop of blood using a small cut in each animal’s tail was obtained to determine the white blood cell (WBCs) differential. The drop was thinly spread over a glass slide and spreading along the edge with a second slide, air dried and stained with the May-Grunewald-Giemsa technique [37]. The absolute number of each type of WBCs was performed by microscope manual techniques counting and categorizing the leukocytes present in six entire fields with 100× objective [38]. Finally, the percentages of differential WBCs were calculated, averaging each kind of WBCs for the mouse blood sample. The basal condition included animals that were not exposed to any treatment.

#### 2.15.4. Peritoneal Macrophage Extraction and Counting

The animals were killed by cervical dislocation, and the macrophages were immediately extracted by injection and 5 mL of PBS into the abdominal cavity and subsequent recovering the same volume later [39]. The cells were pelleted by centrifugation for 5 min at 1500 rpm and suspended in 1 mL of PBS. Finally, the number of macrophages was counted microscopically and the concentration was calculated using a Neubauer chamber.

### 2.16. Statistical Analysis

The values obtained have been expressed as means ± standard deviations (SD) or standard error of the mean (SEM) depending on the case. To determine the statistical differences between each treatment, a one-way analysis of variance (ANOVA) was performed followed by a post-hoc Tukey HSD or Bonferroni’s test, when significant differences were found, considering that the differences were statistically significant for *p* < 0.05. The software used was SigmaPlot version 12.0, 2015 (Systat Software Inc., San Jose, CA, USA).

## 3. Results

### 3.1. Chemical Characterization

#### 3.1.1. Elemental Composition of PcSPs and Biomass and Chemical Composition of Biomass

Elemental analysis of *P. cruentum* was performed for the extracted of PcSPs and the algal biomass. A lower percentage of carbon was observed in polysaccharides than biomass (5.29% and 19.65%, respectively). The percentages of hydrogen were significantly lower for polysaccharides (0.94%) than for biomass (4.01%). The percentage of nitrogen was 0.20% in the polysaccharides and 3.20% in the biomass. Finally, the sulphur content was 0.15% in the polysaccharides and 2.13% in the biomass. Accordingly, the C/N molar ratio was 6.14 in the biomass and 26.72 in the polysaccharides.

Table 1 showed the *P. cruentum* biomass analysis used in the present study (i.e., total proteins, carbohydrates, lipids, inorganic compounds and moisture).

The determinations were carried out in independent experiments in triplicate.

#### 3.1.2. Fourier Transform Infrared Spectroscopy (FT-IR)

The infrared spectra of polysaccharides extracted and purified from *P. cruentum*, ranged from 500 cm^−1^ to 4000 cm^−1^, showing the typical absorption peaks for these substances (Figure 1). In particular, the large absorption peak observed at 3437 cm^−1^ corresponds to the O–H stretching vibration, the weak absorption peaks observed at 2918 cm^−1^ correspond to C-H stretching vibrations. The weak peak at 1739 cm^−1^ indicates a stretching vibration of the carbonyl group in carboxylic acid groups (C=O), and the moderate strength absorption peak at around 1626 cm^−1^ can be associated with a C=O asymmetric stretching vibration, which suggests that *P. cruentum* polysaccharides may contain acetyl or uronic acid groups. The absorption peak at 1468 cm^−1^ likely relates to a C-O symmetric stretching vibration, while the band at 1257 cm^−1^ appears to correspond to asymmetric O=S=O stretching vibration of sulfate esters. Additionally, the peak observed at ca. 1040 cm^−1^ can be associated with glycosidic linkage stretch vibrations of C-O-C bonds, and the weak small absorption peak at 815 cm^−1^ (rather than at 890 cm^−1^) suggests an α-type (instead of a β-type) glycosidic linkage of the polysaccharides. These FT-IR results obtained from *P. cruentum* are entirely in agreement with what would be expected from PsSPs composition.>

#### 3.1.3. Gas Chromatography–Mass Spectrometry (GC-MS)

Concerning the monosaccharide TMS (trimethylsilyl) derivatives composition, Arb (48.3%) was the major component, followed by Gal (23.3%) and Mann (10.3%). The other five types of monosaccharide TMS derivatives, such as Glc, fucose (Fuc), Xyl, ribose (Rib) and Glc-A, were detected through GC-MS analysis. The two main monosaccharides were Glc and Fuc, accounting for 7.4% and 7.0%, respectively. d-glucuronic acid was identified and accounted for 2.5% of the polysaccharide. Finally, Xyl and Rib amounted to 0.7% and 0.4%, respectively (Table 2).

### 3.2. Biological Assessment

#### 3.2.1. LPS Contamination

The Endosafe^®^ assay reported that the level of lipopolysaccharide (LPS) contamination in our polysaccharide fraction at a concentration of 50 μg mL^−1^ was 86 EU mL^−1^. This parameter was much below the that of restrictive standards of the US Pharmacopeia 32 for radiopharmaceuticals (<175 EU mL^−1^). When referred to the weight of polysaccharides introduced in the analyses, the LPS accounted for 0.0017% (*w*/*w*) of the PcSPs.

#### 3.2.2. Antioxidant Activity (ABTS method) in PcSPs and Biomass

The antioxidant activity of the PcSPs was calculated, obtaining a maximum value of 2.23 ± 0.30 μmol TE g^−1^ DW corresponding to 6.92 ± 0.81% at a concentration of 200 µg mL^−1^. On the other hand, the maximum antioxidant activity observed for the biomass was 10.47 ± 0.11 μmol TE g^−1^ DW corresponding to 29.33 ± 0.31% at a concentration of 200 µg mL^−1^ (Figure 2). The results show that there are significant differences between the antioxidant activity of the PcSPs and biomass.

#### 3.2.3. Cytotoxic Effect in HTC-116, U-937, HL-60, MCF-7, NCI-H460 and HGF-1 Cell Lines

The HTC-116, U-937, HL-60, MCF-7, NCI-H460 and HGF-1 cell lines were treated with different concentrations of polysaccharides from *P. cruentum* (Figure 3A–F). As a negative control, the same cell lines were used without treatment. The results obtained with the MTT assay showed that the tested PcSPs inhibited cell proliferation in a dose-dependent manner. The PcSPs significantly inhibited *(p* < 0.001) the viability of the HCT-116 cell line (Figure 3A). The proliferation of human colorectal carcinoma cells was significantly (*p* < 0.001) reduced by the PcSPs, with no significant differences observed between the 2–10 µg mL^−1^, 5–20 µg mL^−1^, 20–312 µg mL^−1^ and 625–2500 µg mL^−1^ concentrations. The 50% inhibitory concentration (IC_50_) of the PcSPs was 2311.20 µg mL^−1^.

In addition, Figure 3B shows a similar experiment using different doses of PcSPs (0–10,000 µg mL^−1^), in which the degree of cytotoxic effect in human leukemia (U-937) cell lines was determined. The PcSPs significantly affected (*p* < 0.001) the viability of the U-937 cell line; no significant differences were observed between the 1–20 µg mL^−1^ and 20–78 µg mL^−1^ and the IC_50_ of the PcSPs was 1676.74 µg mL^−1^. In the case of cell line HL-60 (Figure 3C), no significant differences between the 1–20 µg mL^−1^ and 39–156 µg mL^−1^ concentrations were observed, and the IC_50_ of the PcSPs was 1089.63 µg mL^−1^.

When the breast tumor cell line (MCF-7) was incubated at different concentrations of PcSPs, a significant cell growth inhibition (*p* < 0.001) was observed (Figure 3D). However, no significant differences were observed between the concentrations 0–5 µg mL^−1^, 5–78 µg mL^−1^ and 39–313 µg mL^−1^. The IC_50_ of the PcSPs was 5498.14 µg mL^−1^.

In the same way the PcSPs were incubated with the NCI-H460 cell line at different concentrations, a significant cell growth inhibition (*p* < 0.001) was observed (Figure 3E). However, no significant differences were observed between the concentrations of 1–20 µg mL^−1^ and 78–156 µg mL^−1^. The IC_50_ of the PcSPs was 2861.49 µg mL^−1^. Finally, when PcSPs were incubated with the NCl-H460 cell line at different concentrations, a significant cell growth inhibition (*p* < 0.001) was observed (Figure 3E). However, no significant differences were observed between the concentrations of 1–20 µg mL^−1^ and 78–156 µg mL^−1^. The IC50 of the PcSPs was 2861.49 µg mL^−1^.

Finally, the results obtained with the MTT assay show that the tested PcSPs inhibit cell proliferation in HGF-1 cell line in a dose-dependent manner, with a significant inhibition of cell growth being observed (*p* < 0.001) (Figure 3F). The IC_50_ value of the PcSPs was 5022.55 µg mL^−1^.

#### 3.2.4. Cell Cycle Analysis by Flow Cytometry in Cell Line HL-60

The results obtained in this assay show that, in the presence of PcSPs concentrations higher than IC_50_ (10 mg mL^−1^), HL-60 cells (Figure 4) are mainly found in the sub G_1_ phase (74.18 ± 2.51%), which demonstrates that PcSPs have cytotoxic effect at higher concentrations of IC_50._ By contrast, at lower concentrations of IC_50_ (0.1 mg mL^−1^) the whole cell cycle is observed (i.e., the different phases Sub G_1_, G_0_/G_1_ and S/G_2_/M), indicating that the cytotoxic effect for HL-60 is less significant at lower concentrations of IC_50_.

#### 3.2.5. Determination of Cytokines (IL-6 and TNF-α)

PcSPs induced an increase in TNF-α and IL-6 production in macrophages cell line RAW 264.7 (Figure 5). We found that the saturation of IL-6 was reached when the PcSPs concentration was 50 μg mL^−1^. In the case of TNF-α, saturation induced was not reached for the different concentrations tested herein.

### 3.3. In Vivo Study

#### 3.3.1. Effects of Treatments on White Blood Cell Differential

The polymorphonuclear leukocyte count, such as neutrophils, eosinophils and basophils, in addition to agranulocytes such as lymphocytes and monocytes, was analyzed (Table 3). Neutrophils count showed reduced values at day 21 of treatment for both VEH and PcSPs group compared with basal measures (*p* < 0.001). Difference was found at day 14 when the values of PcSPs were lower than VEH (*p* < 0.05). All count of eosinophils of VEH treatment at day 7, 14 and 21 were elevated related to basal condition measures (*p* < 0.001, *p* < 0.05 and *p* < 0.001 respectively). Only on day 14 was it observed a decrease in eosinophils count (*p* < 0.05). Regarding basophils, no significant differences were found between days, treatments or with the basal condition.

A granulocytes analysis such as both lymphocytes and monocytes also were analysed. All lymphocytes with VEH treatment at day 7, 14 and 21 showed elevated values related to basal condition measures (*p* < 0.05, *p* < 0.01 and *p* < 0.001, respectively). Comparing PcSPs treatment vs. VEH, an increase in lymphocytes was only observed on day 14 (*p* < 0.01). A reduced monocytes count was shown with VEH treatment at both day 7 and 21 related to basal condition measures (*p* < 0.01 and *p* < 0.05 respectively). However, if PcSPs vs. VEH is compared, a high count of monocytes was found at day 7 (*p* < 0.001).

#### 3.3.2. Effects of Treatments on Amount of Peritoneal Macrophages

An important variation on treatment with PcSPs at the end of treatment was found on peritoneal macrophages count (Figure 6). As expected, the level of macrophages was higher than VEH (*p* < 0.01). The number of macrophages were increased with PcSPs group (*p* < 0.001). Further, LPS plus PcSPs combination also showed significant effect compared with single LPS group (*p* < 0.001) although less than single PcSPs group (*p* < 0.001).

## 4. Discussion

In this study, the amounts of protein, lipids, carbohydrates, inorganic compounds and humidity in the *P. cruentum* biomass differ from those observed by Rebolloso et al., 2000 [40], which may be due to the different cultivation conditions of the microalgae. While the percentages of lipids, inorganic compounds and humidity are similar to those reported by Matos et al., 2016 [41], the protein content is more than twice the value obtained in the present work. Rebolloso et al., 2000 [42] showed that *Porphyridium cruentum* also has a carbohydrate content ranging from 40% to 57% of dry weight, with a prevalence of storage amylopectin- and amylose-polyglucans. *P. cruentum* biomass could therefore be used for nutritional purposes due to the amount and diversity of nutrients computed and the absence of toxic factors [40].

FT-IR is a well-established technique that is generally used to determine the main organic functional groups in the field of structural analysis [32]. The PcSPs FT-IR spectrum showed typical absorption peaks for polysaccharides. Additionally, some of the FT-IR peaks of the polysaccharides from *P. cruentum* coincide with those observed in polysaccharides from other algae [43]. In particular, the C-H peak obtained at 2918 cm^−1^ was also observed in the extracellular polysaccharides from *Graesiella* sp. [44], and that of C=O around 1626 cm^−1^ (carboxylate function) was equally identified in the polysaccharides from *Hypnea spinella* (Gigartinales) and *Halopithys incurva* (Ceramiales) [3]. Furthermore, a peak at 1626 cm^−1^, indicating the presence of acetyl and uronic acid groups, was observed both in the polysaccharides from *P. cruentum* and in those from *Haematococcus pluvialis* [45]. A peak at 1257 cm^−1^, assigned to the asymmetric stretching vibration O=S=O of the sulfate esters, was identified both in the polysaccharides from *P. cruentum* and in the sulfated galactans from *Phacelocarpus peperocarpos* [46]. The presence of PcSPs in the present study is in accordance with the observations from Abdala-Díaz et al., 2010 [3]. Similarly, other authors such as Precival and Foyle, 1979 [47] and Huang et al., 2005, [48] have also described the polysaccharides using mainly FT-IR.

The total elemental analysis is a very important assay because it indicates the total amount of C, N, S. Through the FT-IR it is evidenced that the S is found as sulfate when seeing the vibration of this group in the FT-IR at the wavelength of 1257 cm^−1^. Knowing also the amount of total N and using the factor tables of Lourenço et al., 2004 [29] an estimate of the protein levels was obtained.

The identification of the derivatized samples was performed in a GC-MS system, particularly by comparing their retention times with the standard monosaccharides Glc, Gal, Rha, Fru, Mann, Xyl and Api. The identified monosaccharide units were mainly Arb, α-Gal and Mann. Glc-A was also detected and other minor monosaccharides such as Glc and Fuc. Several factors influence the variation in PcSPs—observed even within the same species—and their functional determinants [49]. For example, the SPs extracted from the *Porphyridium* sp. y *P. purpureum* are composed mainly of Xyl and Gal [18,50]. Similarly, in a study of *P. cruentum* by García et al., 1996, the same monosaccharides were reported [51]. The composition of the extracellular polysaccharide produced by the red alga *P. sordidum* revealed mainly Xyl, Glc, D-and L- Gal, Rha, Mann and Glc-A [52]. However, studies reporting Arb as one of the main components are scarce. Gardeva et al., 2009 identified d-Gal, d-Xyl and l-Arb as major monosaccharides in *P. cruentum* [53]. The monomeric composition of *P. cruentum* has been extensively studied by Geresh [18,19,38,39], Gloaguen et al., 2004 [17] and Patel et al., 2013 [54] using different methods of extraction, degradation polysaccharides, identification techniques and cultivation fractions. Bernaerts et al., 2018 [55], performed an extensive profile from four cell wall fractions (extracellular polymeric substances (EPMS), extracellular polysaccharides (EPS), water-soluble cell wall polysaccharides (wsCWPS) and water-insoluble cell wall polysaccharides (wiCWPS)). In general, these studies identified Xyl, d- and l-Gal, Glc and Glc-A, but with different molar ratios. By contrast, our results reveal a different monomeric composition of the *P. cruentum* (yet under the conditions of cultivation and extraction reported herein), where arabinose is the main component. These results may however justify the observed biological activity of *P. cruentum.* PcSPs are a pull of monosaccharides as has been demonstrated in the analysis of GC-MS.

Antioxidants are substances that prevent the oxidation of cells, in particular by scavenging Reactive Oxygen Species (ROS) such as free radicals [56]. Antioxidants also minimize the oxidative damage by increasing natural defences [57]. According to Rudic et al., 2011 [58], the antioxidant activity observed for *P. cruentum* biomass using the ABTS assay is comparable to that obtained for the aqueous extract of *P. cruentum.* Carotenoids are considered to account for most of the microalgae’s antioxidant capacity. However, other compounds such as enzymes, vitamins, polysaccharides and polyunsaturated fatty acids are also involved [59]. In *P. cruentum,* ubiquinones and sulfated polysaccharides are the main contributors to the antioxidant capacity [60]. As expected, a greater scavenging activity is observed in the biomass since this activity is due not only to polysaccharides but also to other components such as carotenoids, phenols, vitamins, etc. that are not found in PcSPs.

It is well known that the composition, molecular weight, structure, sulfate content and type of monosaccharides, among other characteristics, have a significant influence on the biological activities of polysaccharides [56,61,62]. The sulfated polysaccharides from *P. cruentum* have shown antitumor activity, particularly giving rise to a reduction in cell viability both in vitro and in vivo on Graffi tumor cells at different concentrations [53]. Geresh et al. [63] demonstrated that the high molecular weight over sulfated EPS (exo- or extracellular polysaccharides) from *Porphyridium* inhibited neoplastic mammalian cell growth. In our study, MTT tests were performed on different tumor lines, observing a dose-response relationship. Specifically, the cell viability decreasing with increasing concentration of PcSPs, and with the HL-60 cell line showing the lowest IC_50_ (as demonstrated in the HL-60 cytometry assay). Specifically, the percentage of apoptosis was higher at concentrations above its IC_50_, whereas at concentrations below IC_50_, no such activity was observed. Sun et al. [64] evaluated the antitumor and immunomodulatory activities of different-molecular-weight (MW) EPS from *P. cruentum* by the S180-tumor-bearing mouse model in vivo and peritoneal macrophage activation in vitro. EPSs all showed clear immunomodulation effects and inhibited the implanted S180 tumor’s growth in all doses used. Gardavea et al. [65] studied the cytotoxic properties of polysaccharides from *Dixoniella grisea* and *Porphyridium cruentum* on two permanent human cell lines HeLa (cervical carcinoma) and MCF-7 (breast adenocarcinoma), as well as on primary culture from Graffi myeloid tumor in hamsters. Their investigations indicated that both algal polysaccharides showed high cytotoxic and apoptogenic activities on cancer cells and may be a promising alternative to synthetic substances. Gardeva and co-workers [53] suggested that this antitumor activity could be related to this alga’s immunostimulating properties of this alga, and observed that EPS from *Porphyridium* could be the right candidate as an anticancer agent.

The study of the immunomodulatory activity of algae polysaccharides is extensive [66]. In the case of polysaccharides from *P. cruemtum*, IL-6 production was also stimulated in RAW 264.7. Macrophages regulate innate immunity and adaptive immune responses by producing cytokines such as interleukin1β (IL-1β), IL-6 and TNF-α [67]. These signaling molecules are involved in cellular defense processes such as immune responses and inflammatory reactions. In this study, the results obtained indicate that the polysaccharides from *P. cruemtum* are potent inducers of IL-6 cytokines and, most importantly, of TNF-α. TNF-α plays an important role as a regulator of the inflammatory process and autoimmunity through its receptors (TNFR1 or TNFR2), accessory proteins (receptor-associated proteins) and NF-κB, which regulates the expression of genes involved in inflammation, apoptosis and autoimmunity. TNF exerts different functions in different organs, such as the activation of the production of other mediators such as interleukins 1 to 6 [68].

WBCs, also called leukocytes, are the cells of the immune system involved in protecting the body against both infectious disease and foreign invaders [69]. The different white blood cell types can be classified depending on their structure in granulocytes and agranulocytes. In turn, these categories can be divided into five main types: neutrophils, eosinophils, basophils, lymphocytes and monocytes. We analyze the WBCs blood cell differential in order to see the comportment of leukocytes under the administration of PcSPs. We have seen that neutrophils decrease after the three treatment doses with respect to the baseline condition. After the second dose (day 14) a decrease is seen with respect to the vehicle, which shows the improvement of the local stress response compared to the injection that would be acting as a triggering agent. The eosinophils, responsible for pro-inflammatory functions in the pathogenesis of allergic diseases and immediate hypersensitivity, showed interesting results with PcSPs treatment. The PcSPs could decrease the number of eosinophils in blood with respect to VEH mainly after the first dose (day 7), that is to say, after the first day of contact of the animal with the PcSPs. In addition, there were no differences with the basal state with any of the three doses, which would reflect that PcSPs does not present any antigenic activity capable of trigging a hypersensitivity or allergy response. This can also be justified by the non-variation of the basophil count. Regarding lymphocytes, these were found to be increased in the treatment with VEH proportionally to the doses administered, while with PcSPs an increase was only seen in the after second dose (day 14) with respect to the baseline condition of the animals. This increase in cells whose main function is the regulation of the adaptive immune response, it seems that in this case this kind of response would be occurring in the case of VEH injection but not with PcSPs. Then, it could be concluded that there is no specific adaptive response against PcSPs. If we examine the results of treatment with PcSPs on monocytes, the values remain unchanged until after dose 3 (day 21), where they decrease with respect to the basal condition. Therefore, we can also conclude that PcSPs (*p* < 0.001) does not generate an antigenic response that could lead to the recruitment of phagocytes. These results obtained are a first step to determine the null antigenic capacity of PcSPs and their safety when administered in the peritoneum.

The peritoneum has the unusual anatomic feature of being in constant contact with the peritoneal fluid. This fluid from plasma ultrafiltration contains soluble and cellular constituents. Tissue macrophages arise from bone marrow progenitor cells, circulate in the peripheral blood as monocytes, and ultimately migrate into the peripheral tissue sites to take up residence [70]. Like tissue macrophages at other sites, peritoneal macrophages are the most abundant leukocytes present in a healthy peritoneal cavity. The main function of macrophages is the phagocytosis of pathogens or apoptotic cells and the secretion of cytokines and chemokines that directly influence the immune response [69]. During an acute inflammatory response, many macrophages are recruited from the circulation to survive and subsequently undergo a transient proliferative explosion in situ to repopulate the peritoneum [71]. Related to this and analysing our results, we could consider that macrophages resident population at the end of the treatment study is elevated in all groups. We have found increased several macrophages in all treatments compared to VEH. These increases were minor in LPS compared with both PcSPs and LPS plus PcSPs. A possible explanation for lower values in LPS treatment could be the completed immune response against the pathogen. It is known that in the absence of stimuli that induce them to proliferate or activate, macrophages can only survive for a short period [72], but in the case of PcSPs, the macrophages seem to remain more time in the peritoneum. We believe that this fact could be an advantage against some pathogen entering the peritoneal cavity. Less macrophage was obtained with the LPS plus PcSPs combination than LPS, although still more significant than VEH. Related to this and analysing our results, we could consider that macrophages resident population at the end of the treatment study is elevated in all groups. We have found increased several macrophages in all treatments compared to VEH. These increases were minor in LPS compared with both PcSPs and LPS plus PcSPs. A possible explanation for fewer values in LPS treatment could be the completed immune response against the pathogen.

## 5. Conclusions

From the analysis of the FT-IR spectra of PcSPs, we can conclude that the PcSPs in the present study showed typical polysaccharides absorption bands, mainly α- glucan, as well as absorption peaks typical for polysaccharides extracted from other unicellular microalgae. Furthermore, this study suggests that PcSPs have an immunomodulatory and cytotoxic effect on the tumor lines studied mainly in HL-60. In this study, several tumor cell lines were used to evaluate the cytotoxic effect of PcSPs, in addition to the study of the antioxidant activity of biomass and PcSPs. With the results obtained, we can conclude that the polysaccharides from this alga could potentially be used as a valuable ingredient for nutraceutical products. Due to the novelty of our results, we believe that the potential to improve the immunomodulatory response still has many aspects that should be studied in vivo. Those studies should include the specific response that could be improved against more common pathogens or more complex studies such as cluster of differentiation for the identification of cell surface molecules providing immunophenotyping targets.

## Figures and Tables

**Figure 1 biomolecules-11-00488-f001:**
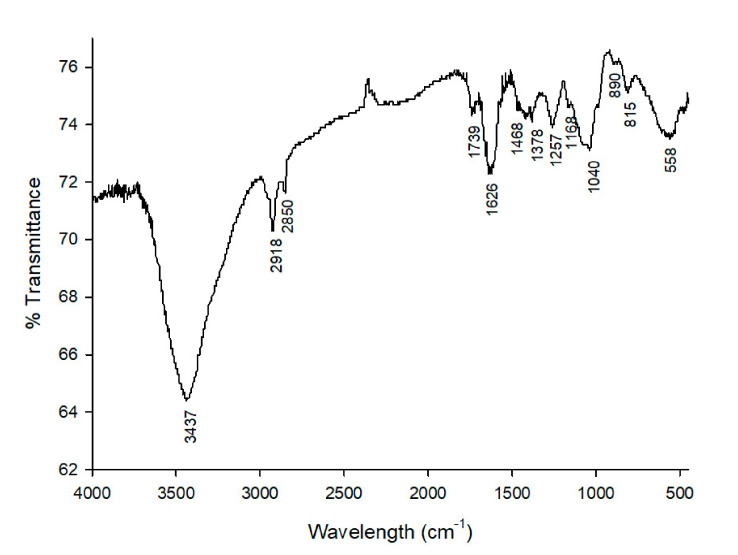
FT-IR spectra of the sulfated polysaccharides (PcSPs).

**Figure 2 biomolecules-11-00488-f002:**
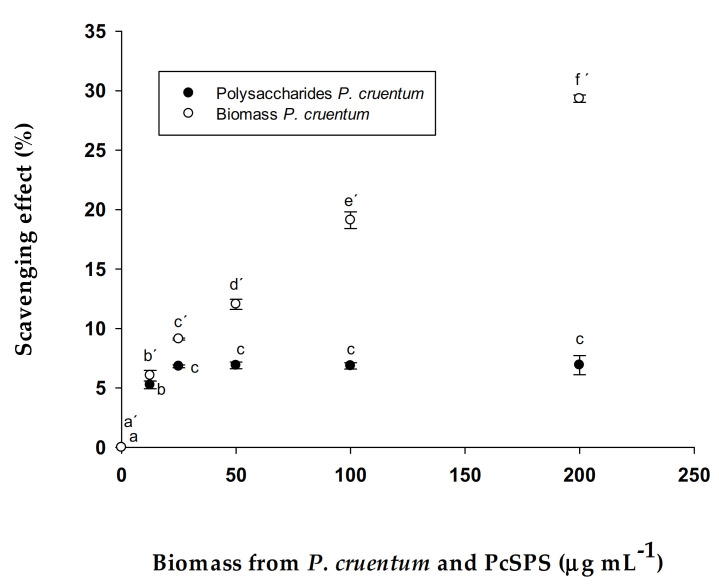
Scavenging effects % of the sample of biomass from *P. cruentum* and PcSPs on ABTS radical. Different points with the same letter correspond to different concentrations with no significant differences between them (post-hoc Tukey HSD test).

**Figure 3 biomolecules-11-00488-f003:**
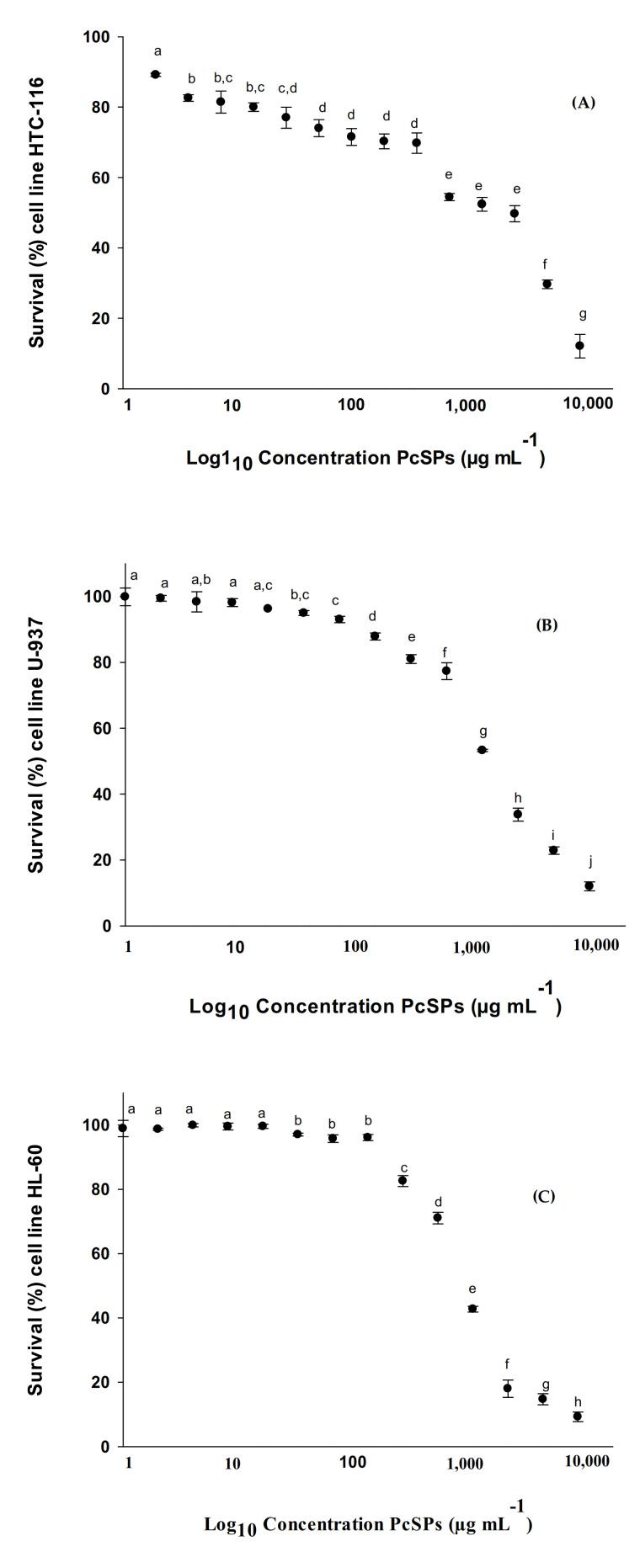
Survival (%) cell lines exposed to different concentrations of PcSPs. (**A**) Survival (%) of HTC-116 cell line; (**B**) Survival (%) of U-937 cell line; (**C**) Survival (%) of HL-60 cell line; (**D**) Survival (%) of MCF-7 cell line, (**E**) Survival (%) of NCI-H460 cell line and (**F**) Survival (%) of HGF-1 cell line. Different points with the same letter correspond to different concentrations with no significant differences between them (post-hoc Tukey HSD test).

**Figure 4 biomolecules-11-00488-f004:**
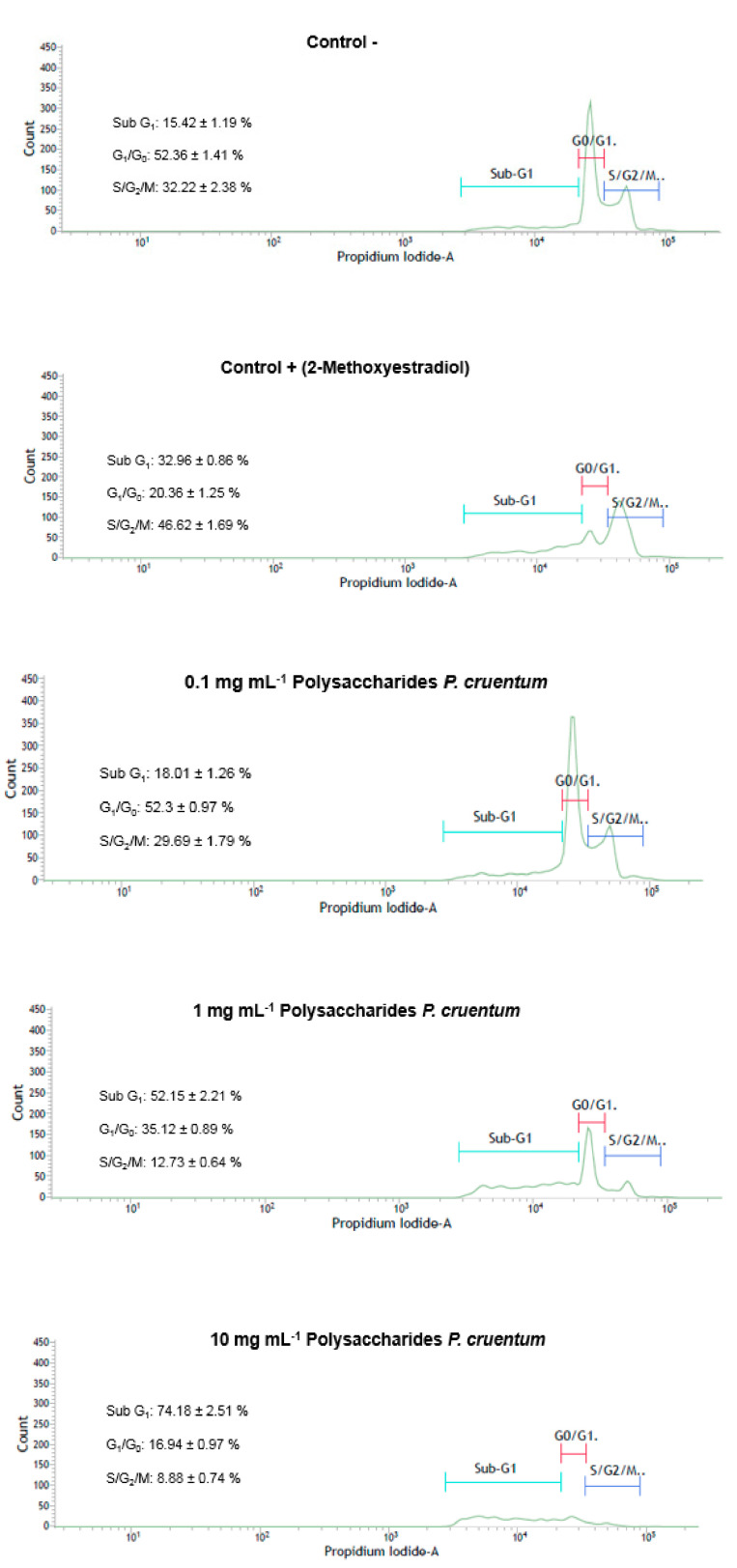
Flow cytometry results for the cell cycle distribution of subpopulations of HL-60 cells in control and samples treated with PcSPs after 24 h. Quantitative analysis of cell cycle data for the full range of tested PcSPs concentration. The plotted values correspond to the mean ± SD from three independent experiments.

**Figure 5 biomolecules-11-00488-f005:**
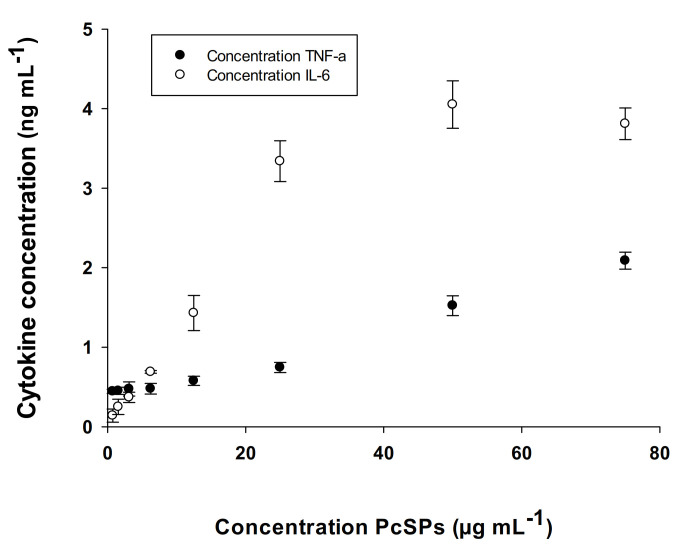
Production of TNF-α and IL-6 by RAW 264.7 exposed to different concentrations of PcSPs. Data points represent the average of eight samples ± SD.

**Figure 6 biomolecules-11-00488-f006:**
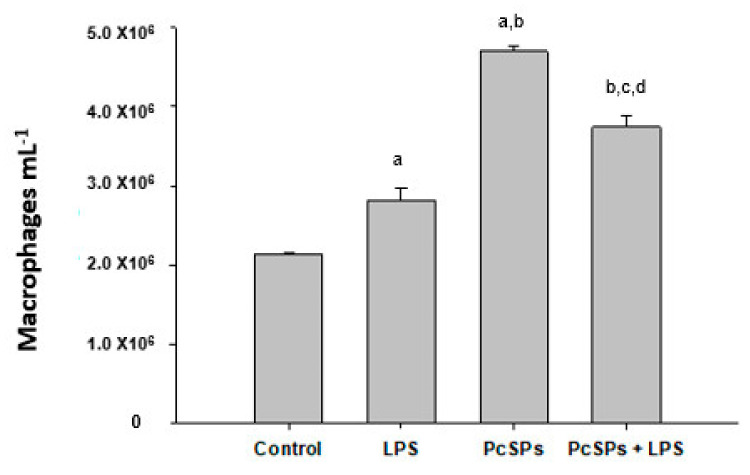
Effects of PcSPs administration on peritoneal murine macrophages counted after treatment. The bars indicate the mean ± SEM (*n* = 6 animals per group). Differences between groups were evaluated using one-way ANOVA and Boferroni’s multiple comparisons test: (a) *p* < 0.01 and (b) *p* < 0.001 vs. CONT group, (c) *p* < 0.001 vs. LPS group and (d) *p* < 0.001 vs. PcSPs group.

**Table 1 biomolecules-11-00488-t001:** Content of proteins, carbohydrates, lipids, ash and moisture in biomass from *P. cruentum* (% of DW) (*n* = 3).

	Percentages
Proteins	15.96 ± 0.09%
Carbohydrates	48.53 ± 0.25%
Lipids	6.85 ± 0.12%
Inorganic compounds	20.14 ± 0.96%
Moisture	8.52 ± 0.19%

Values reported are means ± standard deviations.

**Table 2 biomolecules-11-00488-t002:** Content of monosaccharides in PcSPs.

	Monosaccharide	Retention Time (min)	Peak Area	%
1	Ribose (Rib)	18.42	101,886	0.4
2	Fucose (Fuc)	19.44	1,807,537	7
3	Xylose (Xyl)	20.97	178,746	0.7
4	Arabinose (Arb)	25.46	12,405,336	48.3
5	Mannose (Mann)	26.09	2,650,521	10.3
6–7	α and β-Galactose (Gal)	27.41–27.52	5,994,240	23.3
8	Glucose (Glc)	28.96	1,914,772	7.4
9	Glucuronic acid (Glc-A)	29.83	650,111	2.5

Rib = Ribose; Fuc = Fucose; Xyl = Xylose; Arb = Arabinose; Mann = Mannose; Gal = Galactose; Glc = glucose; Glc-A = Glucuronic Acid.

**Table 3 biomolecules-11-00488-t003:** Effects of PcSPs administration on peritoneal murine macrophages counted after treatment.

WBCs	Basal	Day 7		Day 14		Day 21	
		VEH	PcSPs	VEH	PcSPs	VEH	PcSPs
Neutrophils	22.06 ± 5.33	17.95 ± 6.42	19.29 ± 4.15	22.63 ± 2.88	15.60 ± 4.97 ^Aa^	12.47 ± 4.09 ^c^	11.50 ± 4.31 ^c^
Eosinophils	0.78 ± 0.88	4.40 ± 2.56^c^	1.57 ± 1.40 ^A^	1.50 ± 1.07	0.60 ± 0.52	4.16 ± 0.74 ^a^	5.50 ± 1.78 ^c^
Basophils	0.53 ± 0.61	0.00 ± 0.00	0.00 ± 0.00	0.13 ± 0.35	0.10 ± 0.32	0.00 ± 0.00	0.00 ± 0.00
Lymphocytes	69.89 ± 5.32	76.65 ± 7.34 ^a^	70.57 ± 5.50	63.38 ± 2.00	77.70 ± 5.58 ^Bb^	81.22 ± 4.29 ^c^	82.00 ± 6.86 ^c^
Monocytes	6.56 ± 3.22	1.71 ± 2.21 ^b^	9.83 ± 2.91 ^C^	6.38 ± 1.69	6.00 ± 3.09	2.40 ± 1.52 ^a^	2.83 ± 1.57 ^a^

Values are given as the ± SEM (*n* = 6 animals per group). Differences between groups were evaluated using One-way ANOVA and Boferroni’s multiple comparisons test: (^a^) *p* < 0.05, (^b^) *p* < 0.01 and (^c^) *p* < 0.001 vs. Basal group, (^A^) *p* < 0.05, (^B^) *p* < 0.01 and (^C^) *p* < 0.001 vs. VEH group.

## Data Availability

The data presented in this study are available on request from the corresponding author.

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
