# Peer review of "Immunomodulatory, Antioxidant Activity and Cytotoxic Effect of Sulfated Polysaccharides from Porphyridium cruentum. (S.F.Gray) Nägeli"

_biomolecules, 2021, doi:10.3390/biom11040488_

Round 1
Reviewer 1 Report
The manuscript entitled “Immunomodulatory, antioxidant activity and cytotoxic effect of sulfated polysaccharides from Porphyridium cruentum. (S.F.Gray) Nägeli” by Casas‐Arrojo presents the immunomodulatory, cytotoxic effect and antioxidant activity of the sulfated polysaccharides (PcSPs) isolated from the microalga P. cruentum. The presented results are interesting; however there is a substantial amount of questions that need to be answered before being able to agree with the conclusions reached by the authors. Moreover, the manuscript needs thorough editing. Please see a series of concerns in the following points:
Comments
The various abbreviations should be mentioned the first time the specific term is used in the manuscript and used as is from that point on. Delete the full stops from the titles.
Please check the correct use of superscripts and subscripts throughout the whole document.
The description of the extraction of the polysaccharides would be better to be presented before the 2.3. Total carbon… and 2.4. Biochemical composition… parts.
All the materials used should be included in the Materials and Chemicals section.
The description of the cytotoxicity determination with the MTT assay in all the cell lines used could be presented in the same section and only possible differences could be pointed for the different cell lines.
The results should be presented in a more explanatory and detailed way.
Some results would be better to be presented in the same section (e.g. 3.1.1. and 3.1.2.; 3.2.2. 3.2.4 and 3.2.5.)
Please explain the differences in the percentages of the monosaccharides in Tables 2 and 3. Explain also how the weight was calculated. Has an internal standard been used? Add the information in the respective sections.
The cytotoxic effect of the PcSPs on the different cell lines (cancer, normal, macrophage) should be presented and discussed through the same prism. For example, while the IC50 values for the PcSPs in HTC-116 and RAW 264.7 cell lines are 2311.20 and 2105.02 μg/ml, respectively, in the first case it is stated that PcSPs inhibit cell proliferation, while in the second case, they are non-toxic.
It would be beneficial to confirm the flow cytometry results in another cell line (e.g. the less susceptible to PcSPs)
Please add the respective graphical representation of the survival (%) in RAW 264.7 cell line. It could be referred as Figure 5B.
Comparison of the cytotoxicity results with the respective ones deriving from other known polysaccharides of similar structure is necessary for the significance increase of the conclusions.
The isolation of the polysaccharides from P. cruentum has been previously described by Abdala-Diaz et al., (2010) (Abdala Díaz et al., Ciencias Mar. 2010, 36, 345–353, doi:doi.org/10.7773/cm.v36i4.1732) and the concentration dependence of TNF-α and IL-6 production by RAW 264.7 macrophages has been presented in the same publication (Figures 2 and 3). Please explain the differences, if any. In case Figures 6 and 7 represent new results, they could be presented in one as A and B.
Specific points
Lines 28-29: Please add the exact cell lines used
Line 45: Merge 1st and 2nd paragraphs
Line 100: Delete “and protein content”, since it is described in the next section
Line 120: Please provide the correct name of Cetavlon used
Line 150: Derivatization of polysaccharides, change paragraph and numbering
Line 185: Correct Abs to A in the equation
Line 210: Please correct to five cell lines
Line 232: Please correct to normal cell line
Line 304: Statistical analysis, change paragraph and numbering
Line 376: Please explain a,b,c.etc
Line 435: Please explain a,b,c.etc
Lines 457, 459 and 464: Please correct to measurements
Line 470: Correct to Values are given as the mean value ± SEM
Line 476 and 477: Do you mean LPS by PAT?
Line 489: Merge 1st and 2nd paragraphs
Line 533: Merge 1st and 2nd paragraphs
Line 541: Merge 3rd and 4th paragraphs
Line 567: Please explain briefly why induction of TNF-α is most important.
Author Response
Dear Editor, with regard to the fundamental suggestions made by reviewers, the authors of the manuscript entitled “Immunomodulatory, antioxidant activity and cytotoxic effect of sulfated polysaccharides from Porphyridium cruentum” value each and every one of these.
We are pleased to communicate that we have responded to each one of the requirements.
REVIEW 1:
The manuscript entitled “Immunomodulatory, antioxidant activity and cytotoxic effect of sulfated polysaccharides from Porphyridium cruentum. (S.F.Gray) Nägeli” by Casas‐Arrojo presents the immunomodulatory, cytotoxic effect and antioxidant activity of the sulfated polysaccharides (PcSPs) isolated from the microalga P. cruentum. The presented results are interesting; however there is a substantial amount of questions that need to be answered before being able to agree with the conclusions reached by the authors. Moreover, the manuscript needs thorough editing. Please see a series of concerns in the following points:
POINT 1: The various abbreviations should be mentioned the first time the specific term is used in the manuscript and used as is from that point on. Delete the full stops from the titles
According to what was indicated by the reviewer 1, those terms in which abbreviations have been used have been corrected, indicting their meaning the first time they have been used in the text.
The end points of the titles have been eliminated as indicated by the reviewer 1.
POINT 2: Please check the correct use of superscripts and subscripts throughout the whole document.
As indicated by the reviewer 1, the text has been revised and these errors made throughout the paper have been corrected.
POINT 3: The description of the extraction of the polysaccharides would be better to be presented before the 2.3. Total carbon… and 2.4. Biochemical composition… parts.
We agree with the reviewer and the order of the trials has been changed as indicated.
POINT 4: All the materials used should be included in the Materials and Chemicals section.
We appreciate the reviewer's indication, but we have reviewed the paper and all the compounds used are indicated in the material and methods section where the commercial company is also indicated.
POINT 5: The description of the cytotoxicity determination with the MTT assay in all the cell lines used could be presented in the same section and only possible differences could be pointed for the different cell lines.
We have considered what was indicated by the reviewer and we are in agreement with what was requested so that points 2.11 and 2.13 have been combined in a single point, as reflected in the lines 328-334..
POINT 6: The results should be presented in a more explanatory and detailed way.
We have taken into account the recommendations of the reviewer, and have tried to present the results in a more detailed and explanatory way.
POINT 7: Some results would be better to be presented in the same section (e.g. 3.1.1. and 3.1.2.; 3.2.2. 3.2.4 and 3.2.5.).
Considering the indications of the reviewer, points 3.1.1 and 3.1.2 have been merged into one, leaving the text in the lines 546-580.
In the case of points 3.2.2 and 3.2.4, both sections have also been joined. Leaving the text as indicated in the lines 655-699
In the case of point 3.2.5 it has been eliminated from the paper because this MTT was carried out to see if the concentrations used in the cytokine assay could be used since it is necessary that the concentrations used in this assay have high cell survival.
POINT 8: Please explain the differences in the percentages of the monosaccharides in Tables 2 and 3. Explain also how the weight was calculated. Has an internal standard been used? Add the information in the respective sections.
The authors appreciate the comments.
In section 3.1.4. Gas chromatography–mass spectrometry (GC-MS) Line 615-621 obviously, there is an error, the values (ug) were not determined. Therefore, in section, Table 2 has been modified, eliminating the weight column (ug) and combined with Table 3, incorporating the percentage value (%) that was reported in it. The quantification of sugars was carried out based on the total area of the peaks. These values (%) are those considered for the discussion.
Regarding the use of the standards, these were used for the identification of monosaccharides. Specifically, TMS derivatives were identified by characteristic retention times and mass spectra compared to those of the standards. As well as by comparison of the mass spectra with those of the library of the National Institute of Standards and Technology (NIST 2014).
In section 2.8.2. Gas Chromatography /Mass spectrometry (GC-MS) Analysis the information requested by the reviewer lines 206-218 was incorporated.
POINT 9: The cytotoxic effect of the PcSPs on the different cell lines (cancer, normal, macrophage) should be presented and discussed through the same prism. For example, while the IC50 values for the PcSPs in HTC-116 and RAW 264.7 cell lines are 2311.20 and 2105.02 μg/ml, respectively, in the first case it is stated that PcSPs inhibit cell proliferation, while in the second case, they are non-toxic.
The data obtained from the MTT assay with the RAW 264.7 cell line has been eliminated as this was performed to see which contraction had to be taken to obtain a high survival rate (80%) in order to perform the cytokine determination assay. All cells were compared to normal HGF-1 cells,
POINT 10: It would be beneficial to confirm the flow cytometry results in another cell line (e.g. the less susceptible to PcSPs)
With respect to the reviewer's suggestion, we only performed this test with HL-60 cells as they are cells in suspension and with an adequate size to obtain consistent results with the test that is carried out. U-937s are also cells in suspension but larger in size and the rest of the cells used in the work are adherent cells that form agglomerates, therefore, results are not consistent with this type of assay.
POINT 11: Please add the respective graphical representation of the survival (%) in RAW 264.7 cell line. It could be refered as Figure 5B.
Regarding the reviewer's suggestion, point 3.2.5. has been removed from the paper as this MTT was performed to see if the concentrations used in the cytokine assay could be used since it is necessary that the concentrations used in this assay have high cell survival.
POINT 12: Comparison of the cytotoxicity results with the respective ones deriving from other known polysaccharides of similar structure is necessary for the significance increase of the conclusions.
No more bibliography has been found, with respect to these assays of cell lines with sulfated polysaccharides, than that indicated in the paper.
POINT 13: The isolation of the polysaccharides from P. cruentum has been previously described by Abdala-Diaz et al., (2010) (Abdala Díaz et al., Ciencias Mar. 2010, 36, 345–353, doi:doi.org/10.7773/cm.v36i4.1732) and the concentration dependence of TNF-α and IL-6 production by RAW 264.7 macrophages has been presented in the same publication (Figures 2 and 3). Please explain the differences, if any. In case Figures 6 and 7 represent new results, they could be presented in one as A and B.
This test has corroborated the trend in cytokine synthesis with the results obtained in Abdala et al. 2010.
We join the two figures 6 and 7, as indicated by the reviewer. In the paper it is found as figure 5
Figure 5. Production of TNF-α and IL-6 by RAW 264.7 exposed to different concentrations of PcSPs. Data points represent the average of eight samples ± SD
POINT 14: Lines 28-29: Please add the exact cell lines used
As the reviewer 1 indicates, the cell lines used in the work carried out have been indicated exactly. This can be seen in the lines 27-28.
POINT 15: Line 45: Merge 1st and 2nd paragraphs
As indicated by the reviewer 1, 1st and 2nd paragraphs have been combined.
POINT 16: Line 100: Delete “and protein content”, since it is described in the next section.
Reviewed as requested by reviewer 1.
POINT 17: Line 120: Please provide the correct name of Cetavlon used
As indicated by the reviewer the name of the cetavlon has been corrected “2% Cetylpyridinium bromide hydrate (w/v) (Cetavlon) (Sigma-Aldrich Ref: 285315)” line 137-138
POINT 18: Line 150: Derivatization of polysaccharides, change paragraph and numbering
Reviewed as requested by reviewer 1
POINT 19: Line 185: Correct Abs to A in the equation
Reviewed as requested by reviewer 1, behind line 301.
POINT 20: Line 210: Please correct to five cell lines
They have been corrected as indicated by the reviewer.
POINT 21: Line 232: Please correct to normal cell line
They have been corrected as indicated by the reviewer.
POINT 22: Line 304: Statistical analysis, change paragraph and numbering
Reviewed as requested by reviewer 1, Lines 537-543.
POINT 23: Line 376: Please explain a,b,c…
We appreciate the reviewer's comment, as this had not been included as indicated in the figure. This has been corrected in the explanation of the figure caption “Figure 2. Scavenging effects % of the sample of biomass and PcSPs on ABTS radical. Different points with the same letter correspond to different concentrations with no significant differences between them (post-hoc Tukey HSD test”
POINT 24: Line 435: Please explain a,b,c…
We appreciate the reviewer's comment, as this had not been included as indicated in the figure. This has been corrected in the explanation of the figure caption “Figure 3. Survival (%) cell lines exposed to different concentrations of PcSPs. (A) Survival (%) of cell line HTC-116; (B) Survival (%) of cell line U-937; (C) Survival (%) of cell line HL-60; (D) Survival (%) of cell line MCF-7, (E) Survival (%) of NCI-H460 cell line and (F) Survival (%) of HGF-1 cell line. Different points with the same letter correspond to different concentrations with no significant differences between them (post-hoc Tukey HSD test)”
POINT 25: Lines 457, 459 and 464: Please correct to measurements
Reviewed as requested by reviewer 1.
POINT 26: Line 470: Correct to Values are given as the mean value ± SEM
Reviewed as requested by reviewer 1.
POINT 27: Line 476 and 477: Do you mean LPS by PAT?
Reviewed as requested by reviewer 1.
POINT 27: Line 489: Merge 1st and 2nd paragraphs
Reviewed as requested by reviewer 1.
POINT 28: Line 533: Merge 1st and 2nd paragraphs
Reviewed as requested by reviewer 1.
POINT 29: Line 541: Merge 3rd and 4th paragraphs
Reviewed as requested by reviewer 1.
POINT 30: Line 567: Please explain briefly why induction of TNF-α is most important.
Taking into account the recommendation of the reviewer, the explanation why the induction of TNF-α is more important has been -inserted in the text. Lines 1104-1108.
Reviewer 2 Report
The manuscript submitted by Casas‐Arrojo V et. al. evaluated the immunomodulatory, antioxidant activity and cytotoxic effect of sulfated polysaccharides. The author completed a lot of experimental data in this study. But this MS needs major revision and the author reply comments properly before accepted by Biomolecules. The comments and questions are as follows:
Firstly, in the Introduction part, line 75-78. The author wrote the background in detail. But there was lack of the science question. The introduction should include that question significance.
Line 180, what’s the meaning that “TP” refer? Please explain the abbreviation.
Also, the antioxidant assay, author should add the positive control compared to the samples. Why the author made a comparison for biomass and polysaccharides?
Line 216, cell cycle analysis by flow cytometry that can explain how PcSPs affect the cell proliferation. But why author only choose HL-60 cell to test? Please make a proper description for this assay.
Meanwhile, what’s the purpose for choosing these cell lines? Are they representative cell lines for the experiments?
Line 350-351, “These FT‐IR results are thus from P. cruentum entirely in agreement with what would be expected from PcSPs monosaccharides compositions.” Are there any evidence to support this hypothesis? And how author predicted the monosaccharides composition?
Lastly, line 432-433, there are lack of figures or data to show significant cytotoxic activity of HGF‐1 with respect to MCF‐7 and others.
All in all, after going through the overall manuscript, I suggest authors to make a major revision for this manuscript before publication.
Author Response
Dear Editor, with regard to the fundamental suggestions made by reviewers, the authors of the manuscript entitled “Immunomodulatory, antioxidant activity and cytotoxic effect of sulfated polysaccharides from Porphyridium cruentum” value each and every one of these.
We are pleased to communicate that we have responded to each one of the requirements.
REVIEW 2:
The manuscript submitted by Casas‐Arrojo V et. al. evaluated the immunomodulatory, antioxidant activity and cytotoxic effect of sulfated polysaccharides. The author completed a lot of experimental data in this study. But this MS needs major revision and the author reply comments properly before accepted by Biomolecules. The comments and questions are as follows:
POINT 1: Firstly, in the Introduction part, line 75-78. The author wrote the background in detail. But there was lack of the science question. The introduction should include that question significance.
As indicated by reviewer 2, the science question and the meaning of that question have been included in the introduction. Lines 87-94.
POINT 2: Line 180, what’s the meaning that “TP” refer? Please explain the abbreviation.
I apologize, it was our fault. TP is the Spanish acronym for Phosphate Buffer (PB) in English. This has been corrected in the text. Line 225.
POINT 3: Also, the antioxidant assay, author should add the positive control compared to the samples. Why the author made a comparison for biomass and polysaccharides?
According to the indications of reviewer 2, the positive control used for antioxidant activity is trolox, as indicated in lines 229 and 230.
The comparison was made because it was wanted to see if there was much difference between the activity of the biomass and the polysaccharides at different concentrations.
POINT 4: Line 216, cell cycle analysis by flow cytometry that can explain how PcSPs affect the cell proliferation. But why author only choose HL-60 cell to test? Please make a proper description for this assay.
With respect to the reviewer's suggestion, we only performed this test with HL-60 cells as they are cells in suspension and with an adequate size to obtain consistent results with the test that is carried out. U-937 are also cells in suspension but larger in size and the rest of the cells used in the work are adherent cells that form agglomerates, therefore, results are not consistent with this type of assay.
POINT 5: Meanwhile, what’s the purpose for choosing these cell lines? Are they representative cell lines for the experiments?
The most representative cancers in Spain are colon, breast, leukemia and lung cancer. That is why these cell lines were selected to carry out this study with the PcSPs.
POINT 6: Line 350-351, “These FT‐IR results are thus from P. cruentum entirely in agreement with what would be expected from PcSPs monosaccharides compositions.” Are there any evidence to support this hypothesis? And how author predicted the monosaccharides composition?
One of the most researched representatives of the genus Porphyridium is P. curentum. The author, RT Abdala has worked extensively with this species, so its polysaccharides are components that have already been described by the author (Abdala et al 2010). Similarly, other authors such as; Huang et al 2005, and Precival and Foyle 1979 have also described the polysaccharides using mainly FT-IR. The sentence has been modified in the manuscript, line 994-996. These results are in agreement with those previously reported by Abdala et al 2010 [3], Percival and Foyle 1979 [47] and Huang et al 2005 [48]. References [47] and [48] have been incorporated in the corresponding section line 995.
POINT 7: Lastly, line 432-433, there are lack of figures or data to show significant cytotoxic activity of HGF‐1 with respect to MCF‐7 and others.
We agree with what was indicated by the reviewer, so we have considered that it is better to eliminate this sentence from the paper as it creates confusion. In the graphs corresponding to figure 3 it can be clearly seen how the different concentrations of PcSPs affect the different cell lines used.
All in all, after going through the overall manuscript, I suggest authors to make a major revision for this manuscript before publication.
Reviewer 3 Report
Comments:
Line 26: Correct the sentence.
Line 44-45: this sentence is not necessary.
In few places, grammatical or syntax mistakes are there.
Line 175: Why authors used only ABTS assay? I recommed to include DPPH assay too.
Table 1: How many times authors repeated the experiment?
Line 337: Already this sentence is mentioned in the materials and method. Hence, delete here.
Figure 6, 7: Combine then in one figure via bar diagrams.
Discussion: Need more accurate comparision in few places.
Conclusion: Authors should mention about future recomendation.
Overall, then mansucript is written well.
Author Response
Dear Editor, with regard to the fundamental suggestions made by reviewers, the authors of the manuscript entitled “Immunomodulatory, antioxidant activity and cytotoxic effect of sulfated polysaccharides from Porphyridium cruentum” value each and every one of these.
We are pleased to communicate that we have responded to each one of the requirements.
REVIEW 3:
POINT 1: Line 26: Correct the sentence.
We apologize for the error, it has already been fixed. Lines 23 and 24.
POINT 2: Line 44-45: this sentence is not necessary.
Contrary to what was indicated by reviewer 3, reviewer 1 has indicated that both paragraphs be joined, so that we have finally considered said proposal and both paragraphs have been joined. Lines 41 -.44.
In few places, grammatical or syntax mistakes are there.
POINT 3: Line 175: Why authors used only ABTS assay? I recommed to include DPPH assay too.
The ABTS test was used because it was possible to see how the antioxidant activity varies at different concentrations of biomass and PcSPs.
However, the DPPH assay cannot be used to see the antioxidant activity of PcSPs as they are not soluble in ethanol.
POINT 4: Table 1. How many times authors repeated the experiment?
We appreciate the reviewer's comment as this had not been indicated in the paper previously. With reference to this, it is reported that the tests have been carried out in triplicate each of them. Lines 556.
POINT 5: Line 337: Already this sentence is mentioned in the materials and method. Hence, delete here.
Reviewed and removed as requested by reviewer 3.
POINT 6: Figure 6, 7: Combine then in one figure via bar diagrams.
According to the recommendation of the reviewer 3, which coincides with the recommendation of the reviewer 1. Both figures, 6 and 7, have been combined into one, this being figure 5. However, we have considered that the representation is clearer than the following form. Anyway, we appreciate your indication.
Figure 5. Production of TNF-α and IL-6 by RAW 264.7 exposed to different concentrations of PcSPs. Data points represent the average of eight samples ± SD
POINT 7: Discussion: Need more accurate comparision in few places.
As the reviewer 3 indicates, there has been a more in-depth discussion in some parts of this. Lines 970-1168.
POINT 8: Conclusion: Authors should mention about future recomendation.
We appreciate the reviewer's comment and have included future recommendations that could be made. Lines 1170-1179
Overall, then mansucript is written well.
Round 2
Reviewer 1 Report
Most of the comments have been taken into consideration.
Unfortunately, there was no consistency between the line numbering in the authors’ answers to the reviewers and the manuscript. Additionally, I could not detect some of the indicated by the authors, changes or additions in the manuscript.
Specifically,
Point 6: The results discussion has not been adequately improved.
E.g. What conclusions can be drawn from the elemental analysis and the IR results? Is the fraction PsSPs adequately enriched in polysaccharides? Are the sulfur percentages enough for the polysaccharides to be characterized as sulfated?
What conclusions can be drawn from the scavenging activity results? The antioxidant activity seems to be related to other non polysaccaridic compounds (e.g. carotenoids) as mentioned in the discussion section.
Point 8: I could not see any text added (lines 206-218, as stated by the authors)
Still, there are some corrections to be made.
Lines 383-389. Delete the 4.1.8. Statistical analysis part
Line 399: Please change to Chemical characterization
Line 403: Delete “of sulfated polysaccharides” and the parentheses
Lines 419-424: LPS contatmination part could be transferred to the beginning of the Biological assessment
Line 441: Please change to “results obtained from P. cruentum are entirely in agreement with what would be expected fom PsSPs composition”
Line 449: Change kinds to types
Figure 2: The x-axis title is not correct, (μg mL-1)
Lines 503-505: Please add the cell line to which the results refer.
Line 574: Add parenthesis, )
Line 578: Correct to “showed”
Lines 581, 592, 593: Please correct to “was” instead of were
Line 583: “vs” is not correct as is
Figure 8: Change the numbering to 6
Line 725: Delete “using MI as the internal standard”
Line 727: correct “minority” to “minor”
Line 838: correct “were” to “have”
Line 840: correct “fewer” to “lower”
Line 844: correct “survivor” to “survive”
Line 850: correct “were” to “have”
Line 865: Delete “positively”
Author Response
Dear Editor, with regard to the fundamental suggestions made by reviewers, the authors of the manuscript entitled “Immunomodulatory, antioxidant activity and cytotoxic effect of sulfated polysaccharides from Porphyridium cruentum” value each and every one of these.
We are pleased to communicate that we have responded to each one of the requirements.
Comments and Suggestions for Authors:
Most of the comments have been taken into consideration.
POINT 1: Unfortunately, there was no consistency between the line numbering in the authors’ answers to the reviewers and the manuscript. Additionally, I could not detect some of the indicated by the authors, changes or additions in the manuscript.
We are sorry for the inconvenience that this has caused in the correction of the paper once the appropriate corrections have been made. It must have been a misinterpretation on our part when listing the lines as they were listed without accepting the changes and with this listing the responses to the reviewers were indicated. In the case of accepting the changes, the enumeration varied, so this enumeration would not correspond to the one indicated in the responses to the reviewers. I am sorry that this confusion has occurred. The enumeration of the lines in the article will be done without accepting the changes and therefore they will be indicated in the responses to the reviewer.
POINT 2: Point 6: The results discussion has not been adequately improved.
E.g. What conclusions can be drawn from the elemental analysis and the IR results? Is the fraction PsSPs adequately enriched in polysaccharides? Are the sulfur percentages enough for the polysaccharides to be characterized as sulfated?
As indicated by the reviewer, these questions have been answered in the paper on the lines 1291-1295, 1318-1319 and 1431-1434.
What conclusions can be drawn from the scavenging activity results? The antioxidant activity seems to be related to other non polysaccaridic compounds (e.g. carotenoids) as mentioned in the discussion section.
We have considered what was indicated by the reviewer and your question has been answered in the lines 1328-1331.
POINT 3: Point 8: I could not see any text added (lines 206-218, as stated by the authors)
We agree with the reviewer, one of the questions he posed was not answered correctly in the previous review. We consider that this time the answer is already on line 309-310.
Still, there are some corrections to be made.
POINT 4: Lines 383-389. Delete the 4.1.8. Statistical analysis part.
We agree with the reviewer, and this part was repeated at the point of statistical analysis, so it has been removed from the text leaving it only at the point 2.16. "Statistical analysis".
POINT 5: Line 399: Please change to Chemical characterization.
Reviewed as requested by reviewer 1
POINT 6: Line 403: Delete “of sulfated polysaccharides” and the parentheses.
We appreciate the reviewer's input, but we have reviewed the paper at the indicated point and this was already removed in the first review.
POINT 7: Lines 419-424: LPS contatmination part could be transferred to the beginning of the Biological assessment
We appreciate the reviewer's indications and the LPS contaminationn point has been changed to Biological asseeement.
POINT 8: Line 441: Please change to “results obtained from P. cruentum are entirely in agreement with what would be expected fom PsSPs composition”
Reviewed as requested by reviewer 1
POINT 9: Line 449: Change kinds to types
Reviewed as requested by reviewer 1
POINT 10: Figure 2: The x-axis title is not correct, (μg mL-1)
As indicated by the reviewer 1, the title of the x-axis has been changed to “Biomass from P. cruentum and PcSPs (µg mL-1)”
POINT 11: Lines 503-505: Please add the cell line to which the results refer.
POINT 12: Line 574: Add parenthesis, )
Reviewed as requested by reviewer 1
POINT 13: Line 578: Correct to “showed”
Reviewed as requested by reviewer 1
POINT 14: Lines 581, 592, 593: Please correct to “was” instead of were
Reviewed as requested by reviewer 1
POINT 15: Line 583: “vs” is not correct as is
As the reviewer 1 indicates, "vs" has been changed to "and" in the line indicated by the reviewer.
POINT 16: Figure 8: Change the numbering to 6
Reviewed as requested by reviewer 1
POINT 17: Line 725: Delete “using MI as the internal standard”
Reviewed as requested by reviewer 1
POINT 18: Line 727: correct “minority” to “minor”
Reviewed as requested by reviewer 1
POINT 19: Line 838: correct “were” to “have”
Reviewed as requested by reviewer 1
POINT 20: Line 840: correct “fewer” to “lower”
Reviewed as requested by reviewer 1
POINT 21: Line 844: correct “survivor” to “survive”
Reviewed as requested by reviewer 1
POINT 22: Line 850: correct “were” to “have”
Reviewed as requested by reviewer 1
POINT 23: Line 865: Delete “positively”
Reviewed as requested by reviewer 1
Reviewer 2 Report
The manuscript was revised adequately and can be accepted for publication in Biomolecules.
Author Response
We thank you for your time and for your excellent recommendations to improve the article.